# Anti-inflammatory effects of elexacaftor/tezacaftor/ivacaftor in adults with cystic fibrosis heterozygous for F508del

Heledd H. Jarosz-Griffiths[1]*, Lindsey Gillgrass[2], Laura R. Caley[1], Giulia Spoletini[1,2], Ian J. Clifton[2], Christine Etherington[2], Sinisa Savic[3], Michael F. McDermott[3], Daniel Peckham[1,2]*

1 Leeds Institute of Medical Research, University of Leeds, Leeds, United Kingdom, 2 Adult Cystic Fibrosis Unit, St James's University Hospital, Leeds, United Kingdom, 3 Leeds Institute of Rheumatic and Musculoskeletal Medicine, University of Leeds, Leeds, United Kingdom

* D.G.Peckham@leeds.ac.uk (DP); H.H.Griffiths@leeds.ac.uk (HHJG)

**Data Availability Statement:** Data is within the manuscript and a file has also been uploaded.

## Abstract

Inflammation is a key driver in the pathogenesis of cystic fibrosis (CF). We assessed the effectiveness of elexacaftor/tezacaftor/ivacaftor (ETI) therapy on downregulating systemic and immune cell-derived inflammatory cytokines. We also monitored the impact of ETI therapy on clinical outcome. Adults with CF, heterozygous for F508del (n = 19), were assessed at baseline, one month and three months following ETI therapy, and clinical outcomes were measured, including sweat chloride, lung function, weight, neutrophil count and C-reactive protein (CRP). Cytokine quantifications were measured in serum and following stimulation of peripheral blood mononuclear cells (PBMCs) with lipopolysaccharide (LPS) and adenosine triphosphate and analysed using LEGEND plex™ Human Inflammation Panel 1 by flow cytometry (n = 19). ASC specks were measured in serum and caspase-1 activity and mRNA levels determined from stimulated PBMCs were determined. Patients remained stable over the study period. ETI therapy resulted in decreased sweat chloride concentrations (p < 0.0001), CRP (p = 0.0112) and neutrophil count (p = 0.0216) and increased percent predicted forced expiratory volume (ppFEV1) (p = 0.0399) from baseline to three months, alongside a trend increase in weight. Three months of ETI significantly decreased IL-18 (p< 0.0011, p < 0.0001), IL-1β (p<0.0013, p = 0.0476), IL-6 (p = 0.0109, p = 0.0216) and TNF (p = 0.0028, p = 0.0033) levels in CF serum and following PBMCs stimulation respectively. The corresponding mRNA levels were also found to be reduced in stimulated PBMCs, as well as reduced ASC specks and caspase-1 levels, indicative of NLRP3-mediated production of pro-inflammatory cytokines, IL-1β and IL-18. While ETI therapy is highly effective at reducing sweat chloride and improving lung function, it also displays potent anti-inflammatory properties, which are likely to contribute to improved long-term clinical outcomes.

**Funding:** This work was supported by Leeds Hospitals Charity, Cystic Fibrosis Trust Strategic Research Centre grant (SRC009) The funders had no role in study design, data collection and analysis, decision to publish, or preparation of the manuscript.

**Competing interests:** Conflict of Interest DP: speaker/board honoraria from Vertex This does not alter our adherence to PLOS ONE policies on sharing data and materials.

## Introduction

Cystic fibrosis (CF) is caused by mutations in the cystic fibrosis transmembrane conductance regulator (CFTR) gene which is situated on the long arm of chromosome 7 [1]. The gene encodes for the CFTR protein, an anion channel which conducts chloride and bicarbonate and regulates epithelial sodium transport [2]. In CF, the defective gene results in abnormalities in the production and function of CFTR throughout the body, leading to a multisystem disease affecting many organs including the lungs, pancreas, and gastrointestinal tract.

An absent or defective CFTR results in airway surface liquid acidification and dehydration, abnormal tenacious mucus production and impaired mucociliary clearance [3]. In addition, CF lung disease is associated with exaggerated endobronchial neutrophilic inflammation with an elevation in a wide range of inflammatory biomarkers including neutrophil elastase (NE), IL-8, TNF-$\alpha$, IL-1$\beta$, IL-6 and calprotectin [4]. This persistent inflammatory response is a major contributor of lung damage, even in the context of appropriate and aggressive antibiotic therapy [5]. Studies in infants have demonstrated that the presence of inflammation occurs in the early stages of disease, with increased levels of neutrophils, NE and pro-inflammatory cytokines in broncho alveolar lavage (BAL) fluid being associated with early structural lung damage and disease progression [6–9]. Aberrations in CFTR function also results in a dysregulated cellular response to infection with suboptimal neutrophil antimicrobial activity, increased ER stress, reduced macrophage bacterial killing and a dysregulated interferon response [10–14].

The introduction of ETI therapy has had a profoundly positive effect for the majority of people with CF, with improved lung function, weight and clinical stability [15]. Some of these changes are likely to reflect improved mucociliary clearance, downregulation of inflammation as well as reduced pulmonary exacerbations [16, 17].

There are a growing number of studies supporting the anti-inflammatory effects of ETI therapy, with treatment downregulating airway neutrophilic inflammation, reducing NE activity, increased the percentage of Tregs, and interferon response and reducing lung and serum pro-inflammatory cytokines levels, such as IL-6, IL-8, and IL-17A [11–14, 18–22]. The CF associated inflammatory neutrophil and macrophage phenotypes are also downregulated by ETI therapy resulting in enhanced phagocytosis and intracellular bacterial killing [18, 20, 21, 23–26].

Cystic Fibrosis displays some similarities with systemic autoinflammatory diseases (SAIDs), with CFTR dysfunction and increased ENaC-mediated sodium influx, driving NLRP3-dependent inflammation and an enhanced proinflammatory signature, as evidenced by increased levels of IL-18, IL-1$\beta$, caspase-1 activity and ASC speck release in monocytes and epithelia as well as serum with CF-associated mutations [3, 27, 28]. This exaggerated NLRP3 innate immune response has been shown to be differentially downregulated by tezacaftor/ivacaftor (tez/iva) and lumacaftor/ivacaftor (lum/iva) [27].

The aim of this study was to assess the impact of ETI therapy on clinical status and innate-immune cell-derived cytokine signature in the serum as well as peripheral blood mononuclear cells (PBMCs) harvested from adults with CF (awCF) heterozygous for F508del.

## Materials and methods

### Patients

Consecutive awCF who were heterozygous for F508del were prospectively recruited, between September 2020 and February 2021, from the Leeds Regional Adult CF Unit prior to starting ETI therapy (n = 19). All subjects were naive to previous CFTR modulator therapy. Clinical data were recorded and extracted using the unit's electronic patient records (EMIS web) [29].

All subjects were clinically stable, and lung function, weight, body mass index (BMI), CRP, neutrophil count, serum alt and serum bilirubin were measured at baseline, and again at one and three months of ETI treatment. A sweat test was undertaken before and after three months treatment with ETI and blood samples were collected pre-ETI therapy, and at one and three months, to measure serum and PBMC cytokines levels using our previously described protocol [27]. Healthy controls (HC) were recruited from the Department of Respiratory Medicine and Research laboratories at the Wellcome Trust Benner Building at St James's Hospital. Informed written consent was obtained from all participants at the time of the sample collection. Venous blood was taken at the same time points to assay serum and PBMC cytokine levels. The study was approved by Yorkshire and The Humber Research Ethics Committee (17/YH/0084).

## Samples

The subject's bloods was drawn, using Vacuette tubes (Greiner-Bio-One) containing serum clot activator gel or EDTA for whole blood. Blood samples in the serum clot activator tubes were allowed to clot for 60 min, then centrifuged at 1000 x g for 10 minutes. The resulting serum was then transferred into 1 mL tubes for storage at −80˚C.

## Stimulated cytokine production in PBMCs

PBMCs were extracted from whole blood, using Lymphoprep gradient media (Axis-Shield, Dundee, UK). These cells were then cultured in complete RPMI medium, which consisted of RPMI medium supplemented with 10% heat-inactivated fetal bovine serum, 2 mM L-glutamine, 50 U/ mL penicillin, 50 μg/mL streptomycin. After allowing PBMCs ($2 \times 10^6$/ mL) to settle overnight, they were immediately stimulated with Ultrapure Escherichia coli K12 (Invivogen) LPS (10 ng/ mL, 4 hr), and ATP (5 mM) for 30 min), as previously described [27].

## Cytokine quantification

The amount of cytokines secreted by PBMCs was determined by analysis of the supernatants using the LEGENDplex™ Human Inflammation Panel 1 (13-plex) (BioLegend). The assay was performed in a V-bottom plate according to the manufacturer's protocol and data acquisition was done with the Flow Cytometer—Cytoflex S with High Throughput System (Beckham Coulter). BioLegend's LEGENDplex Data Analysis Software was applied for analysis (www.biolegend.com/legendplex).

## Caspase-1 activity assay

Caspase-2 activity was assessed using a colorimetric assay (Caspase-1 Colorimetrix Assay, R and D Systems, Abingdon, UK), which detects the cleavage of a caspase-specific peptide conjugated to a colour reporter molecule, p-nitroalinine (pNA), performed on cell lysates. The protein concentrations in the lysates were quantified using the Pierce bicinchoninic acid (BCA) assay (Thermo Fisher Scientific, Loughborough, UK).

## Detection of NLRP3-derived ASC specks

For detection of NLRP3 derived ASC specks, 2 μL of PE-conjugated ASC antibody (HASC-71 clone, BioLegend) was added to 100 μL of cell culture media, and the mixture incubated in FACS collection tubes on a shaker for 1 h. Size gating was carried out with Megamix-Plus beads (Biocytex), according to the manufacturer's specifications, and was used to threshold out readings below 0.9 μm. Samples were run and analysed on a Cytoflex-S (Beckman

Coulter). PE-stained particles were determined to be ASC specks in unstimulated and NLRP3 inflammasome stimulated samples. See [30] for detailed methods.

### Detection of mRNA by RT-qPCR

Total RNA isolation was carried out using TRIzol Reagent and Phasemaker tubes (Thermo Fisher Scientific), according to the manufacturer's protocol. RNA quantity and purity were determined using the 260/280 and 260/230 ratios, as determined using a NanoDrop 1000 spectrophotometer (Thermo Fisher Scientific). One hundred nanograms of RNA was converted to cDNA using High-Capacity cDNA reverse Transcription Kit (ThermoFisher Scientific), according to the manufacturer's specifications. Gene expression was analysed using TaqMan probes for NLRP3 (Hs00918082_m1), TXNIP (Hs01006900_g1), TNF (Hs00174128_m1), IL-18 (Hs01038788_m1), pro-IL1B (Hs01555410_m1), IL-10 (Hs00961622_m1), IL-6 (Hs00174131_m1) and HPRT1 (Hs02800695_m1) (Thermo Fisher Scientific) in a reaction containing 100 ng of cDNA, 1 μL TaqMan gene expression assay, 5 μL TaqMan gene expression Master Mix (Thermo Fisher Scientific) and nuclease-free water to a final volume of 10 μL. Cycle parameters were 50˚C for 2 min then 95˚C for 10 min, followed by 40 cycles of 95˚C for 15 sec and 60˚C for 1 min; samples were processed in duplicate on a QuantStudio 7 Flex Real-Time PCR System (Applied Biosystems). Data were expressed as relative expression compared to the housekeeping gene, HPRT1.

### Analysis and statistics

Throughout the manuscript, the baseline (pre-therapy, zero month) values for each patient were calculated as a percentage of the average baseline within the patient group receiving ETI, as per our previous publication investigating double modulator therapy [27]. This explains why the starting values do not all congregate at 100% in the graphs. The one month and three-month samples are calculated as a percentage of the baseline average. A non-parametric Kruskal-Wallis statistical test, with Dunn's multiple comparison test was performed to assess changes in clinical and biochemical parameters. A one-way ANOVA statistical test, with Tukey's multiple comparison test was performed to assess cytokine and ASC speck responses following ETI therapy. A p value of $< 0.05$ was considered significant.

## Results

### Clinical response in patients receiving ETI therapy

Nineteen adults, median (range) age of 31 (18–63) years, were consecutively recruited prior to starting ETI therapy. All but one was Caucasian. Treatment resulted in a significantly decreased median sweat chloride (p<0.0001), CRP (p = 0.0112), ppFEV$_1$ (p = 0.0399) and neutrophil counts (p = 0.0216), from baseline to three months, alongside a trend increase in weight (Table 1).

### Serum cytokine responses in patients receiving ETI therapy

ETI reduced NLRP3 inflammasome activity, this was evidenced by a significant reduction in serum IL-1β (p = 0.0013) cytokine levels at 3 months and a significant reduction in and IL-18 (p<0.0001) and NLRP3-inflammasome associated ASC specks at 1 month and 3 months post ETI therapy (Fig 1A, 1B and 1H). ETI therapy reduced NLRP3-mediated inflammation by 16 pg/ml for IL-1β, and by 49 pg/ml for IL-18 at 3 months post ETI therapy. Although reduced relative to baseline, the average levels remained higher than that of healthy control (HC) volunteers' sera. Of the 19 pwCF, n = 7 had IL-18 cytokine levels, reduced to HC levels or less;

**Table 1. Demographic and clinical characteristics for 19 adults with CF heterozygous for F508del CFTR mutations receiving ETI.** Data are expressed as median and range.

| | elexacaftor / tezacaftor/ ivacaftor (ETI) | | |
|---|---|---|---|
| Mutations (n) | F508del/R347P (2) <br> F508del/621+1G->T (1) <br> F508del/G85E (1) <br> F508del/ c.2052dupA p.Gln685Thrfs*4 (1) <br> F508del/4326delTC (1) <br> F508del/G542X (1) <br> F508del/N1303K (1) <br> F508del/3659delC (3) <br> F508del/c.1219G>T p.Glu407Ter (1) <br> F508del/I507del (1) <br> F508del/1154insTC (1) <br> F508del/2184delA (2) <br> F508del/1461ins4 (1) <br> F508del/V456A (1) <br> F508del/c.4028dupG p.Cys1344LeufsTer15 (1) | | |
| Female (n) | 11 (19) | | |
| Age, years | 27 (17–63) | | |
| Days on IV previous 12 months | 14 (0–89) | | |
| Days on IV following 12 months | 0 (0–14) | | |
| Chronic Pseudomonas (n) | 7 (19) | | |
| Intermittent Pseudomonas | 8 (19) | | |
| No Pseudomonas | 3 (19) | | |
| Burkholderia cepacia complex (Genomovar II) | 1(19) | | |
| Month of Rx | 0 | 1 | 3 |
| | Median (range) | | |
| ppFEV$_1$ | 66 (44–91) | 76.5 (45–124) | 77.5 (53–118) |
| ppFVC | 83.5 (60–111) | 95 (67–129) | 92 (63–119) |
| BMI | 22.63 (19.1–43.7) | 23.75 (19.8–25.8) | 24.29 (19.5–47.9) |
| CRP (mg/ L) | 5 (5–52) | 5 (5–7.6) | 5.0 (5–5) |
| Sweat chloride (mmol/ L) | 103 (64–131) | | 56 (16–85) |
| Neutrophil count | 5.65 (2.42–12.89) | 4.17 (2.08–6.35) | 3.655 (2.52–8.26) |
| Serum Alt | 25 (15–49) | 24 (10–80) | 22.5 (10–60) |
| Serum bilirubin | 7 (3–11) | 10.5 (4–12) | 9 (3–32) |

n = 4 for IL-1β, and n = 11 for NLRP3-inflammasome associated ASC specks (Fig 1A and 1B, blue lines; S1 Fig). There was no significant correlations with the clinical metadata, as such no clinical predictions could be made to identify pwCF who would normalise by 3 months. An ETI-mediated reduction was also observed in NLRP3-inflammasome independent cytokines, with significant reductions observed in TNF ($p = 0.0028$) and IL-6 ($p = 0.0109$) levels, following three month's treatment with ETI therapy (Fig 1C and 1D). No changes were observed in IL-10, IL-8, or IL-23 levels (Fig 1E–1G).

## NLRP3-stimulated cytokine production in PBMCs

Next, we investigated cytokine differences at a cellular level, using PBMCs isolated from CF patients and HC volunteers. Baseline (pre-therapy) CF-PBMCs stimulated with LPS and ATP, and HC-PBMCs (across three months) were compared (Fig 2 and S2 Fig). There was a significant reduction in IL-1β ($p = 0.0476$), IL-18 ($p<0.0001$) cytokine levels, as well as a reduction in caspase-1 activity ($p = 0.0128$) at three months consistent with NLRP3-mediated activation,

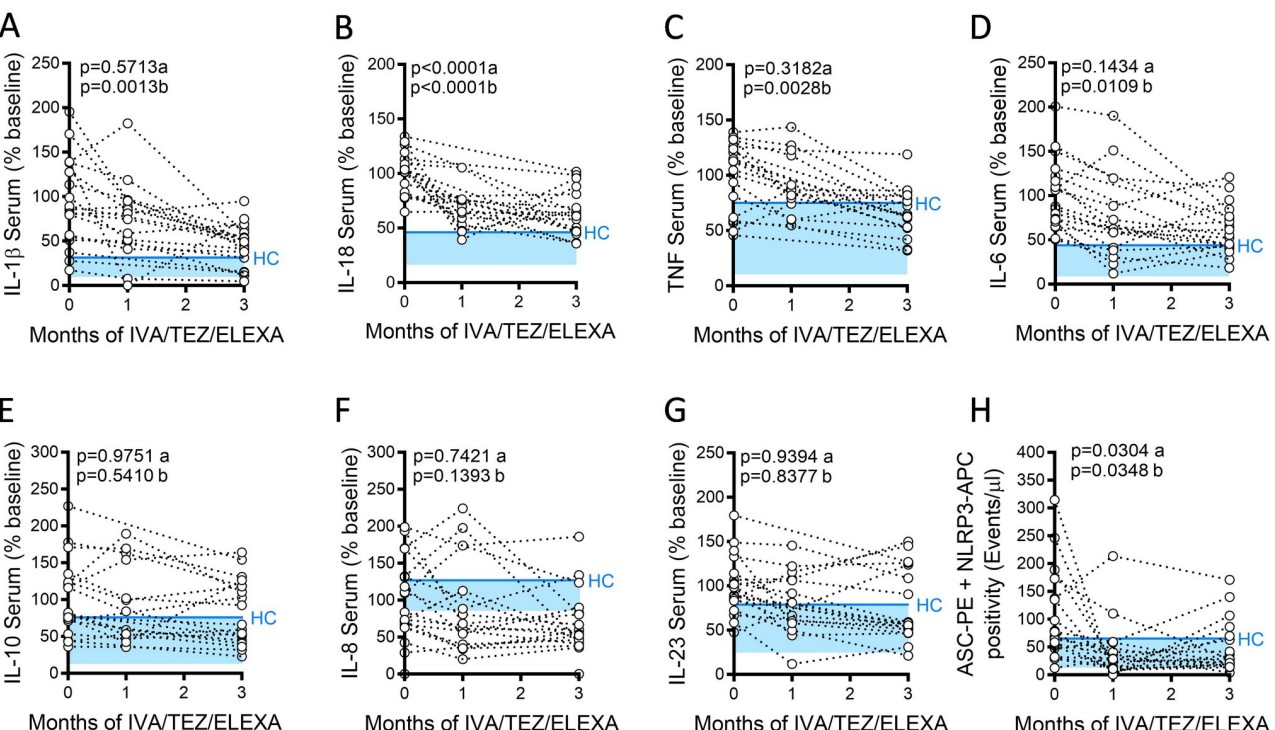

**Fig 1. Serum cytokine levels in patients with CF (heterozygous, F508del, other), following treatment with ETI.** Sera were collected at baseline, one month and three months of treatment from patients heterozygous for F508del, and a second minimal function CFTR mutations receiving ETI therapy (n = 19). LegendPlex assays were used to detect levels of A, IL-18; B, IL-1β; C, TNF; D, IL-6, E, IL-10, F IL-8, G IL-23 in sera. H, Flow cytometry was used to detect NLRP3-positive ASC-specks. A one-way ANOVA statistical test, with Tukey's multiple comparison test was performed. P values for baseline to one month (a) and baseline to three months (b) are shown on each graph. The upper 95% confidence interval for average HC (solid blue line) across 3 months, with block colour shading to lower 5% confidence interval, is displayed for each cytokine.

as demonstrated in the sera samples (Fig 2A, 2B and 2H). ETI therapy reduced the NLRP3-mediated inflammation by 773 pg/ml for IL-1β, and by 119 pg/ml for IL-18. Although reduced relative to baseline, the average levels remained higher than that of healthy control volunteer stimulated PBMCs (Fig 2A and 2B, blue lines; S2 Fig). Consistent with the sera, NLRP3-independent cytokines, TNF (p = 0.0033) and IL-6 (p = 0.0216), were also significantly reduced following ETI therapy, with changes for IL-8 (p = 0.0784) and IL-23 (p = 0.0615) approaching significance. There were also near significant increases in IL-10 following one month of ETI treatment (p = 0.0268) but falling at three months (p = 0.9525).

## mRNA expression in NLRP3-inflammasome activated PBMCs

The corresponding mRNA levels for pro-IL-1β, IL-18, TNF, IL-6, and IL-10 were monitored across three months. At three months post-ETI, a significant decrease in mRNA levels of pro-IL-1β (p = 0.0013), IL-18 (p = 0.0013) and TNF (p = 0.0101) and near significant decrease for IL-6 (p = 0.0682) (Fig 3A–3D) were observed. There were no significant differences in IL-10, NLRP3 and TXNIP transcript levels, from baseline to three months (Fig 3, S3 Fig).

## Discussion

This study evaluated the clinical and anti-inflammatory effect of ETI in awCF who were modulator naïve and heterozygous for F508del. Treatment resulted in significant clinical

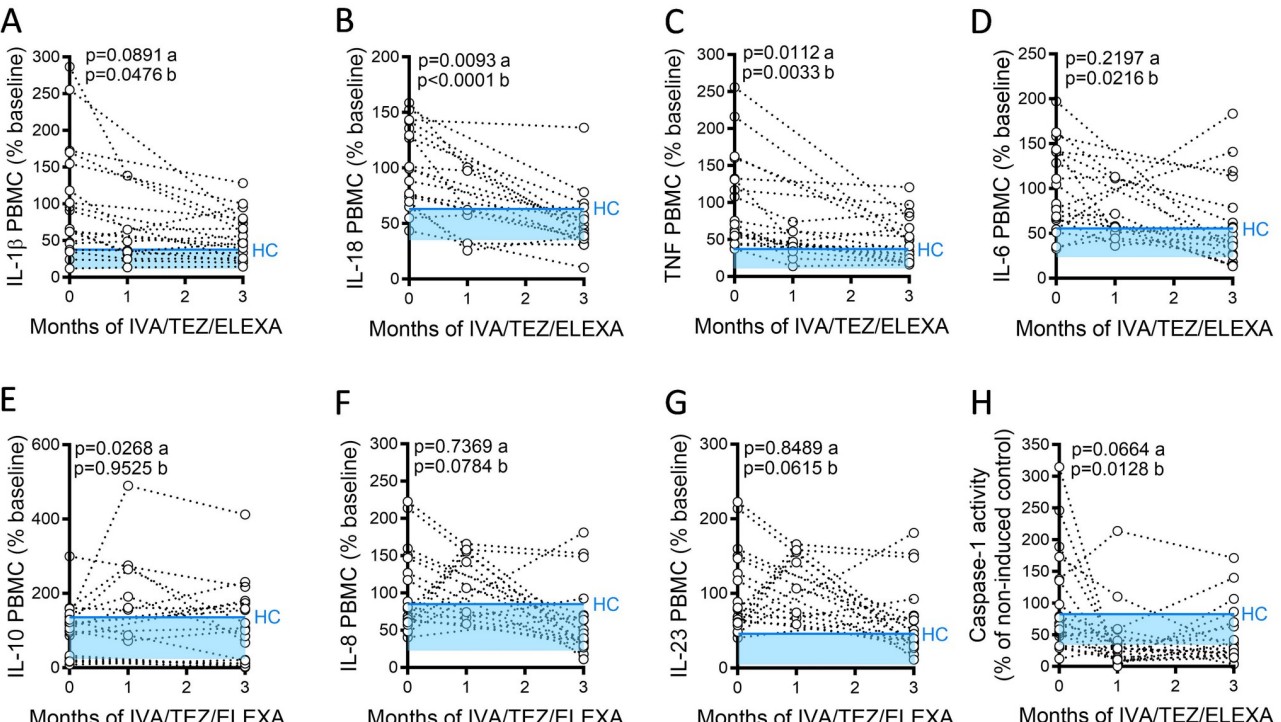

**Fig 2. Cytokine secretion in NLRP3-stimulated CF immune cells isolated from patients with CF (heterozygous, F508del, other), following treatment with ETI.** PBMCs isolated at baseline, one month and three months from patients heterozygous, F508del, other CFTR mutations receiving ETI therapy (n = 19). Following isolation, PBMCs were immediately stimulated with LPS (10 ng/mL, 4 hr), and ATP (5 mM) for the final 30 min. Legend Plex assays were used to detect levels of A, IL-1β; B, IL-18; C, TNF; D, IL-6 and E, IL-10, F IL-8, G, IL-23 secretion from PBMCs. H, caspase-1 activity was detected in stimulated PBMCs at each time point. A two-way ANOVA statistical test with Tukey's multiple comparison test was performed. P values for baseline to one month (a) and baseline to three months (b) are shown on each graph. The upper 95% confidence interval for average HC (solid blue line) across 3 months, with block colour shading to lower 5% confidence interval is displayed for each cytokine.

improvement, reduced sweat chloride concentrations, downregulation of routine serum inflammatory markers as well as a significant reduction in both CF serum and stimulated PBMCs IL-18, IL-1β, TNF and IL-6 levels.

We have previously shown that serum and PBMCs from awCF exhibit an enhanced NLRP3-inflammasome signature that is downregulated differentially by by tez/iva and lum/iva [3, 27]. While IL-18 and TNF levels were found to decrease significantly following three months of treatment, IL-6 levels remained unchanged and IL-1β only declined following iva/tez [27].

In contrast, ETI appears to induce a more rapid and potent anti-inflammatory response when compared to tez/Iva and lum/Iva [27]. Treatment was associated with downregulated of the NLRP3 inflammasome response in LPS/ATP stimulated PBMCs, with a significant reduction in NLRP3 positive ASC-specks and caspase 1. Expression of pro-IL-1β, IL-18 and TNF mRNA was also significantly downregulated with a reduction in NLRP3 mRNA between one and three months, and a trend towards a decrease in IL-6 mRNA levels. A transient increases in IL-10 occurred at 1 month in both serum and post PBMC stimulation. This may reflect temporary normalisation of a reduced IL-10 response, which has been previously reported in CF macrophages [31]. The number of subjects recruited into this study were limited by the availability of modulator eligible, naive subjects, who were heterozygous for F508del, as the majority of patients had started treatment. This may have resulted in a cohort of more stable patients

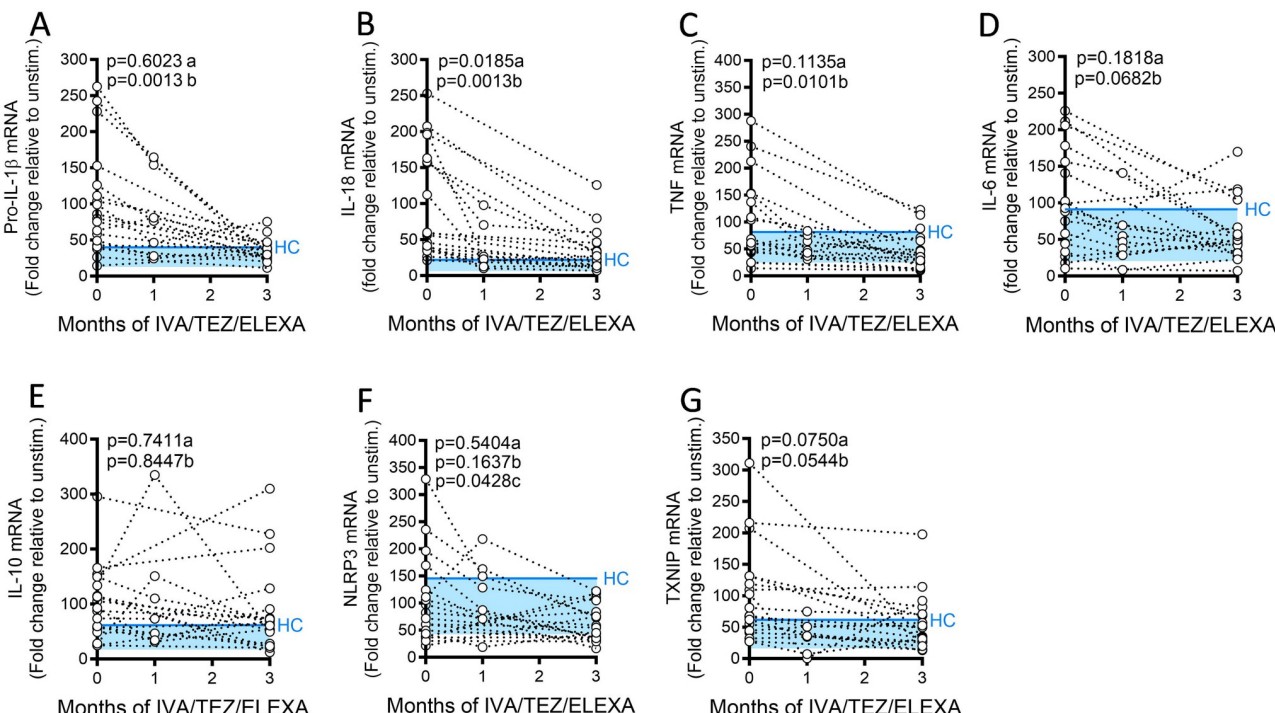

**Fig 3. mRNA expression in NLRP3-stimulated CF immune cells isolated from patients with CF (heterozygous, F508del, other), following treatment with ETI.** PBMCs isolated at baseline, one month and three months from patients heterozygous, F508del, other CFTR mutations receiving ETI therapy (n = 19). Following isolation, PBMCs were immediately stimulated with LPS (10 ng/mL, 4 hr), and ATP (5 mM) for the final 30 min. RNA was isolated, and mRNA gene expression levels measured A, IL-1β; B, IL-18; C, TNF; D, IL-6 and E, IL-10, F NLRP3, G, TXNIP. A two-way ANOVA statistical test with Tukey's multiple comparison was performed. P values for baseline to one month (a) and baseline to three months (b) are shown on each graph. The upper 95% confidence interval for average HC (solid blue line) across 3 months, with block colour shading to lower 5% confidence interval is displayed for each cytokine.

and influenced the level of inflammatory markers. For instance, IL-8 and IL-23 levels remained unchanged between baselines, one and three months. While serum IL-8 levels have been reported to fall post ETI, the change is not always significant and measurement have been previously taken after a more prolonged treatment period [19, 32]. Changes in IL-8 levels as well as elevation in IL-23 in CF sputum appear to be more marked with IL-6, being potentially, a more sensitive circulating biomarker [20].

A more recent study by Gabillard-Lefor and colleagues, using a similar three month protcol, demonstrated downregulation of inflammasome activation in CF monocytes following ETI, with a reduction in IL-1β and caspase activity [28]. The authors identified P2X7R overexpression as a driver of inflammasome activation with ETI therapy reducing ATP/P2X7R induced inflammasome activation [28]. The majority of subjects were F508del homozygouse (70%), 65% had previously been been on modualtors and recrutment had been prioritised to individuals with severe lung disease [28].

There are several limitations of this study, including the relatively small sample size and the short duration of treatment. While the study design replicated previous published work on the different anti-inflammatory effect of double modulator therapy, a longer-term response would have provided invaluable data on the level and persistence of the anti-inflammatory response. However, large prospective studies, such as PROMISE and RECOVER, are ongoing and are likely to shed further light on the longer-term impact of ETI therapy on both lung infection and inflammation [33, 34]. While the study was not designed or powered to investigate the

clinical efficacy of ETI treatment, we were reassured by the significant clinical improvement, which mirrored larger clinical trials. We did not see any correlation between sweat chloride, lung function and cytokine response as might be expected. While the overall level of changes in sweat chloride concentration can reflect clinical outcome in cohort studies, it does not necessarily reflect an individual response to modulator therapy [35]. Other factors such as genotype, disease phenotype and variation in ETI metabolism may also affect variation in molecular and inflammatory responses. The numbers of subjects in this study were too small to characterise differences and predicators of response between individuals, such as genotype.

In conclusion, while ETI is highly effective at reducing sweat chloride and improving lung function, it also displays potent anti-inflammatory properties, which are likely to contribute to considerably improved long-term clinical outcomes.

## Supporting information

**S1 Fig. Serum cytokine levels in healthy control volunteers over 3 months (0, 1 and 3 month time points).** Sera were collected at zero-month, one month and three month from HC (n = 10). LegendPlex assays were used to detect levels of A, IL-18; B, IL-1β; C, TNF; D, IL-6; E, IL-10; F, IL-8; G, IL-23 in serum. H, Flow cytometry was used to detect NLRP3 positive ASC-specks. A one-way ANOVA statistical test with Tukey's multiple comparison was performed. P value for baseline to one month (a) and baseline to three months (b) shown on each graph. (TIF)

**S2 Fig. Cytokine secretion in NLRP3-stimualted immune cells isolated from healthy control volunteers over 3 months (0, 1 and 3 month time points).** Sera were collected at zero-month, one month and three month from HC volunteers (n = 10). Following isolation, PBMCs were immediately stimulated with LPS (10ng/mL, 4hr), and ATP (5mM) for the final 30 min. LegendPlex assays were used to detect levels of A, IL-1β; B, IL-18; C, TNF; D, IL-6; E, IL-10; F, IL-8; G, IL-23 secretion from PBMCs. A Caspase-1 activity was detected in stimulated PBMCs at each time point. A two-way ANOVA statistical test with Tukey's multiple comparison was performed. P value for baseline to one month (a) and baseline to three months (b) shown on each graph. (TIF)

**S3 Fig. mRNA expression in NLRP3-stimualted immune cells isolated from healthy control volunteers over 3 months (0, 1 and 3 month time points).** Sera were collected at zero-month, one month and three month from HC volunteers (n = 10). Following isolation, PBMCs were immediately stimulated with LPS (10ng/mL, 4hr), and ATP (5mM) for the final 30 min. RNA was isolated, and mRNA gene expression levels measured A, IL-1β; B, IL-18; C, TNF; D, IL-6; E, IL-10; F, NLRP3; G, TXNIP. A two-way ANOVA statistical test with Tukey's multiple comparison was performed. P value for baseline to one month (a) and baseline to three months (b) shown on each graph. (TIF)

**S1 Data.**
(XLSX)

## Author Contributions

**Conceptualization:** Heledd H. Jarosz-Griffiths, Daniel Peckham.

**Data curation:** Heledd H. Jarosz-Griffiths, Lindsey Gillgrass, Daniel Peckham.

**Formal analysis:** Heledd H. Jarosz-Griffiths.

**Funding acquisition:** Heledd H. Jarosz-Griffiths, Ian J. Clifton, Sinisa Savic, Michael F. McDermott, Daniel Peckham.

**Investigation:** Heledd H. Jarosz-Griffiths, Lindsey Gillgrass, Laura R. Caley, Giulia Spoletini, Ian J. Clifton, Christine Etherington, Sinisa Savic, Michael F. McDermott, Daniel Peckham.

**Methodology:** Heledd H. Jarosz-Griffiths, Sinisa Savic, Michael F. McDermott, Daniel Peckham.

**Project administration:** Heledd H. Jarosz-Griffiths, Daniel Peckham.

**Resources:** Heledd H. Jarosz-Griffiths, Daniel Peckham.

**Supervision:** Daniel Peckham.

**Writing – original draft:** Heledd H. Jarosz-Griffiths, Laura R. Caley, Michael F. McDermott, Daniel Peckham.

**Writing – review & editing:** Heledd H. Jarosz-Griffiths, Lindsey Gillgrass, Laura R. Caley, Giulia Spoletini, Ian J. Clifton, Christine Etherington, Sinisa Savic, Michael F. McDermott, Daniel Peckham.

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
