## [Decision Letter · Decision Letter 0]

26 Mar 2024

PONE-D-24-06228

Anti-inflammatory effects of elexacaftor/tezacaftor/ivacaftor in adults with cystic fibrosis heterozygous for F508del

PLOS ONE

Dear Dr. Peckham,

Thank you for submitting your manuscript to PLOS ONE. After careful consideration, we feel that it has merit but does not fully meet PLOS ONE’s publication criteria as it currently stands. Therefore, we invite you to submit a revised version of the manuscript that addresses the points raised during the review process.

Although the points raised by reviewers were relatively minor, it is critical that authors carefully address each of the comments from reviewers as follows: 

Modify and adequate the introduction focusing on inflammation on CF.Make sure to include and discuss any clinical metadata (e.g. genotype, sweat chloride values, etc.), as suggested by Reviewer 1.Please include study limitations in the discussion and highlight similarities or discrepancies of your data, as compared with recent publications in the field.

We look forward to receiving your revised manuscript.

Kind regards,

Santiago Partida-Sanchez

Academic Editor

PLOS ONE

Journal Requirements:

When submitting your revision, we need you to address these additional requirements. 1. Please ensure that your manuscript meets PLOS ONE's style requirements, including those for file naming. The PLOS ONE style templates can be found at https://journals.plos.org/plosone/s/file?id=wjVg/PLOSOne_formatting_sample_main_body.pdf and https://journals.plos.org/plosone/s/file?id=ba62/PLOSOne_formatting_sample_title_authors_affiliations.pdf 2. We noticed you have some minor occurrence of overlapping text with the following previous publication(s), which needs to be addressed: -https://doi.org/10.7554/eLife.54556-https://doi.org/10.1016/S1569-1993(22)00366-6 In your revision ensure you cite all your sources (including your own works), and quote or rephrase any duplicated text outside the methods section. Further consideration is dependent on these concerns being addressed. 3. Please note that funding information should not appear in any section or other areas of your manuscript. We will only publish funding information present in the Funding Statement section of the online submission form. Please remove any funding-related text from the manuscript. 4. Thank you for stating the following financial disclosure:    "This work was supported by Leeds Hospitals Charity, Cystic Fibrosis Trust Strategic Research Centre grant (SRC009)" Please state what role the funders took in the study.  If the funders had no role, please state: "The funders had no role in study design, data collection and analysis, decision to publish, or preparation of the manuscript." If this statement is not correct you must amend it as needed. Please include this amended Role of Funder statement in your cover letter; we will change the online submission form on your behalf. 5. Thank you for stating the following in the Competing Interests section:    "Conflict of InterestDP: speaker/board honoraria from Vertex
" Please confirm that this does not alter your adherence to all PLOS ONE policies on sharing data and materials, by including the following statement: "This does not alter our adherence to  PLOS ONE policies on sharing data and materials.” (as detailed online in our guide for authors http://journals.plos.org/plosone/s/competing-interests).  If there are restrictions on sharing of data and/or materials, please state these. Please note that we cannot proceed with consideration of your article until this information has been declared.  Please include your updated Competing Interests statement in your cover letter; we will change the online submission form on your behalf. 6. When completing the data availability statement of the submission form, you indicated that you will make your data available on acceptance. We strongly recommend all authors decide on a data sharing plan before acceptance, as the process can be lengthy and hold up publication timelines. Please note that, though access restrictions are acceptable now, your entire data will need to be made freely accessible if your manuscript is accepted for publication. This policy applies to all data except where public deposition would breach compliance with the protocol approved by your research ethics board. If you are unable to adhere to our open data policy, please kindly revise your statement to explain your reasoning and we will seek the editor's input on an exemption. Please be assured that, once you have provided your new statement, the assessment of your exemption will not hold up the peer review process.

Reviewers' comments:

Reviewer's Responses to Questions

**Comments to the Author**

1. Is the manuscript technically sound, and do the data support the conclusions?

Reviewer #1: Yes

Reviewer #2: Yes

2. Has the statistical analysis been performed appropriately and rigorously? 

Reviewer #1: Yes

Reviewer #2: Yes

3. Have the authors made all data underlying the findings in their manuscript fully available?

Reviewer #1: Yes

Reviewer #2: Yes

4. Is the manuscript presented in an intelligible fashion and written in standard English?

Reviewer #1: Yes

Reviewer #2: Yes

5. Review Comments to the Author

Reviewer #1: This is a straight-forward manuscript that adds to growing evidence of partial effects of ETI on inflammatory responses in CF. While the findings are not necessarily novel and studies are limited, it is still of interest to the community and contains relevant human data.

Minor

-due to the ever increasing diagnostic incidence of CF across the globe, I would remove the first sentence “Cystic fibrosis (CF) is one of the most common life limiting autosomal recessive diseases to affect populations of western European descent”.

-legend for table 1 says n of 20, while everything says 19 subjects

- the time points in the serum cytokine responses paragraph should be made more clear as IL-18 was significant starting at 1 month, but everything else was only significant at 3 months. I would also comment on the proportion of individuals that had values equal or less than healthy controls by 3 months. Were there any clinical metadata (such as genotype) that predicted who would normalize by 3 months and those that remained elevated above controls?

-were the same individuals from figure 1 consistent in figure 2 in terms of those with persistently elevated cytokine secretion vs those with normalized secretion?

-it is interesting that IL-8 serum levels and secretion appears to be below non-CF for many individuals at baseline and maintains this way. Any explanation? Based on second CFTR variant?

-why does Fig 3 not completely match Figs1 and 2? IL-8, IL-23 not in Fig 3

- are these all White awCF? Or any race/ethnic heterogeneity?

-is there any correlation with changes in sweat chloride and cytokine responses?

Reviewer #2: Very interesting paper on the effects of ETI on inflammation. It's an important and very current topic. The paper is useful but requires additions and modifications. Furthermore, the case studies are very limited to provide conclusions.

Major comments:

- introduction and discussion are limited. The authors need to talk more about inflammation and CF, explain the reasons that make the study useful and necessary. The initial part of the discussion is more of an introduction.

- some cytokine values are reduced post ETI, others are not. The authors must explain or at least assume the reasons

- the limits of the paper are totally missing, starting from the very limited case series.

- the discussion is very limited. The authors should compare their results with other papers. There are international projects on post-ETI inflammation, such as PROMISE. Why don't the authors talk about it?

6. PLOS authors have the option to publish the peer review history of their article (what does this mean?). If published, this will include your full peer review and any attached files.

Reviewer #1: No

Reviewer #2: No

---

## [Author Response · Author response to Decision Letter 0]

9 May 2024

Title: Anti-inflammatory effects of elexacaftor/tezacaftor/ivacaftor in adults with cystic fibrosis heterozygous for F508del

PLOS ONE

We are delighted to submit a corrected version of our manuscript for consideration for publication in PLOS ONE. We are very grateful to the reviewers for their constructive and in depth feedback. We have responded to each point raised by yourself and the reviewers and uploaded a marked-up copy of our manuscript that highlights changes made to the original version as well as a separate file without track changes. We have also included all data in a separate file as requested. 

We have summarised all the changes bellow. Page and line relate to marked up copy

Yours sincerely

Response to reviewers

Editor

Modify and adequate the introduction focusing on inflammation on CF. 

Author Response

We are grateful for this comment. We had initially focused more on NLRP3 but recognise that a more in depth discussion would improve the article. We have therefore re written the introduction with a broader focus on inflammation. 

Page 3 line 65 onwards

In addition, CF lung disease is associated with exaggerated endobronchial neutrophilic inflammation with an elevation in a wide range of inflammatory biomarkers including neutrophil elastase (NE), IL-8, TNF-α, IL-1β, IL-6 and calprotectin [4]. This persistent inflammatory response is a major contributor of lung damage, even in the context of appropriate and aggressive antibiotic therapy [5]. Studies in infants have demonstrated that the presence of inflammation occurs in the early stages of disease, with increased levels of neutrophils, NE and pro-inflammatory cytokines in broncho alveolar lavage (BAL) fluid being associated with early structural lung damage and disease progression [6-9]. Aberrations in CFTR function also results in a dysregulated cellular response to infection with suboptimal neutrophil antimicrobial activity, increased ER stress, reduced macrophage bacterial killing and a dysregulated interferon response[10-14]. 

The introduction of ETI therapy has had a profoundly positive effect for the majority of people with CF, with improved lung function, weight and clinical stability[15]. Some of these changes are likely to reflect improved mucociliary clearance, downregulation of inflammation as well as reduced pulmonary exacerbations[16, 17]. 

There are a growing number of studies supporting the anti-inflammatory effects of ETI therapy, with treatment downregulating airway neutrophilic inflammation, reducing NE activity, increased the percentage of Tregs, and interferon response and reducing lung and serum pro-inflammatory cytokines levels, such as IL-6, IL-8, and IL-17A [11-14, 18-22]. The CF associated inflammatory neutrophil and macrophage phenotypes are also downregulated by ETI therapy resulting in enhanced phagocytosis and intracellular bacterial killing [18, 20, 21, 23-26]. 

Cystic Fibrosis displays some similarities with systemic autoinflammatory diseases (SAIDs), with CFTR dysfunction and increased ENaC-mediated sodium influx, driving NLRP3-dependent inflammation and an enhanced proinflammatory signature, as evidenced by increased levels of IL-18, IL-1β, caspase-1 activity and ASC speck release in monocytes and epithelia as well as serum with CF-associated mutations[3, 27, 28]. This exaggerated NLRP3 innate immune response has been shown to be differentially downregulated by tezacaftor/ivacaftor (tez/iva) and lumacaftor/ivacaftor (lum/iva)[27].

Editor

Make sure to include and discuss any clinical metadata (e.g. genotype, sweat chloride values, etc.), as suggested by Reviewer 1.

Author’s response

We are grateful for this comment. The numbers of patients in this study were too small to sub analyse the impact of factors such a genotype. In response to this comment, we have altered the discussion as advised to include discussion on clinical metadata. 

Page 14 line 301

While the study was not designed or powered to investigate the clinical efficacy of ETI treatment, we were reassured by the significant clinical improvement, which mirrored a larger clinical trial. We did not see any correlation between sweat chloride, lung function and cytokine response as might be expected. While the overall level of changes in sweat chloride concentration can reflect clinical outcome in cohort studies, it does not necessarily reflect an individual response to modulator therapy. Other factors such as genotype, disease phenotype and variation in ETI metabolism may also affect variation in molecular and inflammatory responses. The numbers of subjects in this study were too small to characterise differences and predicators of response between individuals, such as genotype.

Editor

Please include study limitations in the discussion and highlight similarities or discrepancies of your data, as compared with recent publications in the field

.

Author’s response

We are grateful for this comment. We have included a section on limitations in the discussion and highlighted similarities and discrepancies with recent publications. 

Page 15 line 309

A transient increases in IL-10 occurred at 1 month in both serum and post PBMC stimulation. This may reflect temporary normalisation of a reduced IL-10 response which has been previously reported in CF macrophages [31]. 

The number of subjects recruited into this study were limited by the availability of modulator eligible, naive subjects, who heterozygous for F508del as the majority of patients had started treatment. This may have resulted in a cohort of more stable patients and influenced the level of inflammatory markers. For instance, IL-8 and IL-23 levels remained unchanged between baseline, one and three months. While serum IL-8 levels have been reported to fall post ETI, the change is not always significant and measurement have been previously taken after a more prolonged treatment period [19, 32]. Changes in IL-8 levels as well as elevation in IL-23 in CF sputum appear to be more marked with IL-6, being potentially, a more sensitive circulating biomarker [20]. 

Page 13 line 289

There are several limitations of this study, including the relative small sample size and the short duration of treatment. While the study design replicated previous published work on the different anti-inflammatory effect of double modulator therapy, a longer-term response would have provided invaluable data on the level and persistence of the anti-inflammatory response. However, large prospective studies, such as PROMISE and RECOVER, are ongoing and are likely to shed further light on the longer-term impact of ETI therapy on both lung infection and inflammation [33, 34]. While the study was not designed or powered to investigate the clinical efficacy of ETI treatment, we were reassured by the significant clinical improvement, which mirrored larger clinical trial. 

Editor 

While revising your submission, please upload your figure files to the Preflight Analysis and Conversion Engine (PACE) digital diagnostic tool, https://pacev2.apexcovantage.com/. PACE helps ensure that figures meet PLOS requirements. 

Author’s response

We have uploaded all images as requested through PACE and all files met PLOS requirement

Journal Requirements:

Please ensure that your manuscript meets PLOS ONE's style requirements, including those for file naming. The PLOS ONE style templates 

Authors

We have modified the references according to PLOS guidelines 

Journal Requirements:

We noticed you have some minor occurrence of overlapping text with the following previous publication(s), which needs to be addressed:

Authors

We have addressed these. 

Journal Requirements:

In your revision ensure you cite all your sources (including your own works), and quote or rephrase any duplicated text outside the methods section. Further consideration is dependent on these concerns being addressed.

Authors

We have addressed these.

Journal Requirements:

Please note that funding information should not appear in any section or other areas of your manuscript. We will only publish funding information present in the Funding Statement section of the online submission form. Please remove any funding-related text from the manuscript.

Authors

We have removed funding detail within the manuscript as requested. 

Journal Requirements:

Thank you for stating the following financial disclosure: 

 "This work was supported by Leeds Hospitals Charity, Cystic Fibrosis Trust Strategic Research Centre grant (SRC009)" Please state what role the funders took in the study. 

Authors

We have added, "The funders had no role in study design, data collection and analysis, decision to publish, or preparation of the manuscript." 

Journal Requirements:

Thank you for stating the following in the Competing Interests section: 

 "Conflict of Interest DP: speaker/board honoraria from Vertex "

Author’s response

 "This does not alter our adherence to PLOS ONE policies on sharing data and materials.” 

Author’s response 

We have uploaded the data in a separate file 

Reviewer 1

due to the ever increasing diagnostic incidence of CF across the globe, I would remove the first sentence “Cystic fibrosis (CF) is one of the most common life limiting autosomal recessive diseases to affect populations of western European descent”.

Author’s response: 

This sentence has been removed. 

Reviewer 1

legend for table 1 says n of 20, while everything says 19 subjects

Author’s response: 

We thank the reviewer for picking this up and it has been corrected.

Reviewer 1

The time points in the serum cytokine responses paragraph should be made more clear as IL-18 was significant starting at 1 month, but everything else was only significant at 3 months. I would also comment on the proportion of individuals that had values equal or less than healthy controls by 3 months. Were there any clinical metadata (such as genotype) that predicted who would normalize by 3 months and those that remained elevated above controls?

Author’s response: 

We thank the reviewer for this interesting comment. 

We have added the following sentence to page 15 line 309

A transient increases in IL-10 occurred at 1 month in both serum and post PBMC stimulation. This may reflect temporary normalisation of a reduced IL-10 response which has been previously reported in CF macrophages [31]. 

Page 11 line 207

We have added the proportion of individuals that had values equal or less than healthy controls by 3 months in the result section.

Reviewer 1

Were there any clinical metadata (such as genotype) that predicted who would normalize by 3 months and those that remained elevated above controls?

Author’s response

The numbers were small and not powered to identify such trends. This an important point and we have added the following statement page 17 line 43

While the study was not designed or powered to investigate the clinical efficacy of ETI treatment, we were reassured by the significant clinical improvement, which mirrored larger clinical trial. We did not see any correlation between sweat chloride, lung function and cytokine response as might be expected. While the overall level of changes in sweat chloride concentration can reflect clinical outcome in cohort studies, it does not necessarily reflect an individual response to modulator therapy. Other factors such as genotype, disease phenotype and variation in ETI metabolism may also affect variation in molecular and inflammatory responses. The numbers of subjects in this study were too small to characterise differences and predicators of response between individuals, such as genotype. 

We also included this sentence in the results section:

Page 11 line 209

There was no significant correlations with the clinical metadata, as such no clinical predictions could be made to identify pwCF who would normalise by 3 months.

Reviewer 1

-were the same individuals from figure 1 consistent in figure 2 in terms of those with persistently elevated cytokine secretion vs those with normalized secretion?

Author’s response: 

Overall, there were consistent patterns in cytokine levels between serum and PBMC stimulated cells following ETI treatment. CF162 was the only sample that had inconsistencies across more than one cytokine, with levels of IL-1� and TNF found to be reduced in the serum sample but not in the PBMC stimulated samples, where levels were unchanged. There was also a discrepancy with CF217 in IL-1�, CF209 with IL-18 and CF211 and CF215 with TNF.

Reviewer 1

-it is interesting that IL-8 serum levels and secretion appears to be below non-CF for many individuals at baseline and maintains this way. Any explanation? Based on second CFTR variant?

Author’s response: 

 page 15 line 314

We have expanded on this is the following paragraph 

The number of subjects recruited into this study were limited by the availability of modulator eligible, naive subjects, who heterozygous for F508del as the majority of patients had started treatment. This may have resulted in a cohort of more stable patients and influenced the level of inflammatory markers. For instance, IL-8 and IL-23 levels remained unchanged between baseline, one and three months. While serum IL-8 levels have been reported to fall post ETI, the change is not always significant and measurement have been previously taken after a more prolonged treatment period [19, 32]. Changes in IL-8 levels as well as elevation in IL-23 in CF sputum appear to be more marked with IL-6, being potentially, a more sensitive circulating biomarker [20]. 

Reviewer 1

-why does Fig 3 not completely match Figs1 and 2? IL-8, IL-23 not in Fig 3

- are these all White awCF? Or any race/ethnic heterogeneity?

Author’s response: 

As we did not see any consistent changes in IL-8 and IL-23 levels in PBMCs or serum we decided not to explore changes in mRNA. We chose to look at potential changes in NLRP3 and TXNIP expression on the basis that the changes in IL-18 and IL-1� cytokine levels could potentially be due to changes in NLRP3 expression. TXNIP can directly interact with and activate NLRP3, and can be stimulated by changes in cellular ROS.

Reviewer 1

-is there any correlation with changes in sweat chloride and cytokine responses?

Author’s response: page 17 line 45

We have added the following sentence as previously discussed

We did not see any correlation between sweat chloride, lung function and cytokine response as might be expected. While the overall level of changes in sweat chloride concentration can reflect clinical outcome in cohort studies, it does not necessarily reflect an individual response to modulator therapy.

Reviewer #2: 

- introduction and discussion are limited. The authors need to talk more about inflammation and CF, explain the reasons that make the study useful and necessary. The initial part of the discussion is more of an introduction. 

the discussion is very limited. The authors should compare their results with other papers. There are international projects on post-ETI inflammation, such as PROMISE. Why don't the authors talk about it?

Author’s response

We have changed the introduction and discussion as advised, expanded the overview of inflammation and modulators.

 In addition we have added the following sentence

Page 16 line 3

---

## [Editor Report · Decision Letter 1]

15 May 2024

Anti-inflammatory effects of elexacaftor/tezacaftor/ivacaftor in adults with cystic fibrosis heterozygous for F508del

PONE-D-24-06228R1

Dear Dr. Peckham,

We’re pleased to inform you that your manuscript has been judged scientifically suitable for publication and will be formally accepted for publication once it meets all outstanding technical requirements.

Kind regards,

Santiago Partida-Sanchez

Academic Editor

PLOS ONE
---

## [Editor Report · Acceptance letter]

20 May 2024

PONE-D-24-06228R1 

PLOS ONE

Dear Dr. Peckham, 

I'm pleased to inform you that your manuscript has been deemed suitable for publication in PLOS ONE. Congratulations! Your manuscript is now being handed over to our production team.

Kind regards, 

on behalf of

Professor Santiago Partida-Sanchez 

Academic Editor

PLOS ONE